# SSUL: Semantic Segmentation with Unknown Label for Exemplar-based Class-Incremental Learning

**Sungmin Cha**[1,2]*, **Beomyoung Kim**[3]*, **YoungJoon Yoo**[2,3], **and Taesup Moon**[1]

[1] Department of Electrical and Computer Engineering, Seoul National University
[2] NAVER AI Lab, [3] Face, NAVER Clova
sungmin.cha@snu.ac.kr, {beomyoung.kim, youngjoon.yoo}@navercorp.com,
tsmoon@snu.ac.kr

## Abstract

This paper introduces a solid state-of-the-art baseline for a class-incremental semantic segmentation (CISS) problem. While the recent CISS algorithms utilize variants of the knowledge distillation (KD) technique to tackle the problem, they failed to fully address the critical challenges in CISS causing the catastrophic forgetting; the semantic drift of the background class and the multi-label prediction issue. To better address these challenges, we propose a new method, dubbed SSUL-M (Semantic Segmentation with Unknown Label with Memory), by carefully combining techniques tailored for semantic segmentation. Specifically, we claim three main contributions. (1) defining *unknown* classes within the background class to help to learn future classes (help plasticity), (2) *freezing* backbone network and past classifiers with binary cross-entropy loss and pseudo-labeling to overcome catastrophic forgetting (help stability), and (3) utilizing *tiny exemplar memory* for the first time in CISS to improve both plasticity and stability. The extensively conducted experiments show the effectiveness of our method, achieving significantly better performance than the recent state-of-the-art baselines on the standard benchmark datasets. Furthermore, we justify our contributions with thorough ablation analyses and discuss different natures of the CISS problem compared to the traditional class-incremental learning targeting classification. The official code is available at https://github.com/clovaai/SSUL.

## 1 Introduction

Class incremental learning (CIL) problem, in which a learner should incrementally learn newly arriving class objects while not "catastrophically" forgetting the past learned classes, is one of the fundamental, yet still open, problems in machine learning. After the seminal work, [27], most of the recent neural network-based CIL research has focused on the classification setting, and various approaches have been proposed to address the main challenge of the problem, the so-called plasticity-stability dilemma, *e.g.*, [20, 30, 12, 1, 2, 8], to name a few.

The CIL framework has been extended to more complex semantic segmentation tasks [24, 3, 7], motivated by the practical need in various applications such as autonomous driving. One of the key additional difficulties of the class-incremental semantic segmentation (CISS) problem lies in the semantic drift of the background class present in the incrementally arriving training data. Namely, the label 'background (BG)" is assigned to all the pixels not included in the *current* class object region. The "BG" pixels belong to *three* categories: future object classes that the model does not yet observes, past object classes that are already learned, and the true background.

---

*Equal contribution.

35th Conference on Neural Information Processing Systems (NeurIPS 2021).

A few recent works attempted to address the above semantic drift issue by leveraging and modifying the knowledge-distillation (KD) [20] technique, popular for CIL in standard image classification. Namely, the initiative study [24] modified the KD for CISS straightforwardly, and [3] proposed a strategy to incorporate the BG class probability in computing the cross-entropy and distillation losses. Furthermore, [7], the current state-of-the-art, utilized the pseudo-labeling of the BG pixels of the current task data with the model of the previous task using the cross-entropy loss, and it applies the multi-scale feature distillation scheme adopted from [8]. However, we argue that above works only partially addressed the BG label issue. That is, [3] naively *added* the class probabilities to modify the cross-entropy and distillation losses making it hard to have fine-grained learning of prediction probability for each class, each pixel. Moreover, [7] could only handle the BG class pixels to the *past* classes via pseudo-labeling and lacked any mechanism for handling the *future* class case, one possible option for the the BG class. Consequently, their CIL segmentation performance, measured by mean Intersection-over-Union (mIoU), has been significantly lower than the upper-bound, the case of joint-training with all the labels.

This paper first identifies that the multi-label prediction of semantic segmentation is another critical challenge of CISS and proposes SSUL-M (Semantic Segmentation with Unknown Label with Memory) to address the challenge. Specifically, Our contributions are summarized as follows. First, we introduce an additional "*Unknown*" class label assigned to the objects in the background, detected by the off-the-shelf saliency-map detector. We let the base feature extractor distinguish the representations of the potential future class objects and the actual background region by augmenting the BG label with this additional class. Second, we adopt the pseudo-labeling strategy as in [7] and further augment the BG & Unknown class labels with the past class labels, but with two essential differences in concrete learning strategies to stabilize the classification scores and improve the *precision* of the prediction. One is using the separate *sigmoid*, instead of the softmax, output classifier for each class so that the model can learn the logit score in an absolute sense per class. The other is *freezing* the base feature extractor and the classifiers for past classes after initial learning to strictly maintain the past classes' knowledge. Third, we utilize an *exemplar memory* to store a tiny portion of training data, including past classes, as anchors and further improve the mIoU. Note that using the exemplar memory is a standard practice for CIL in classification but has been overlooked in CISS. Moreover, we show that the memory helps improve the mIoU for the *current* classes, in contrast to a common belief that it is a tool to prevent forgetting of past classes.

By integrating the above contributions, SSUL-M achieved the state-of-the-art performance on popular benchmarks with a *significantly large* margin over the recent baselines [3, 7], particularly when the number of incremental tasks gets larger. Furthermore, we conduct extensive ablation studies and both quantitative and qualitative analyses to convincingly highlight the strength of our method.

## 2   Related Work

**Class Incremental Learning (CIL)**   CIL [28, 25] considers the setting in which new class objects arrive incrementally and the model needs to classify *all* the classes without storing all the past class data. It is well known that neural network-based CIL suffers from catastrophic forgetting [23], caused by the score bias toward the new classes due to the training data imbalance. Most CIL studies have focused on the classification tasks, and the exemplar-memory based methods combined with KD [30, 12, 1, 2, 8] achieved the state-of-the-art performance by re-balancing the biased predictions of the classifier.

**Class Incremental Semantic Segmentation (CISS)**   Contrary to the current trend of CIL for classification, CIL for semantic segmentation have only focused on the setting without utilizing the exemplar-memory [24, 3, 7]. [3] first addressed the semantic drift of BG label, and [7] proposed to use pseudo-labels to augment the BG label, all without exemplar-memory. To our knowledge, we firstly use exemplar-memory and potential future classes in BG label to solve CISS.

**Saliency Map Detection**   Salient object detection is a fundamental computer vision task that identifies the most visually distinctive objects in an image. We use the off-the-shelf saliency-map detector to define the *Unknown* class in the BG label. The early salient object detection method, DRFI [13], conducts a multi-level segmentation with a random forest regressor to detect a salient object. Recent deep neural network based method exploits rich multi-scale feature maps with short

connections [11] and pooling-based modules [21]. The saliency information has widely been utilized on various tasks. For example, the weakly-supervised semantic segmentation approaches [29, 31, 14] generate pseudo-labels filtering out the background regions using saliency map, and recent data augmentations [17, 16] also exploit the saliency information to find the optimal mixing of mask.

## 3 Proposed Method

### 3.1 Notations and Problem Setting

We consider exactly the same setting as considered in [3, 7]. In CISS, the learning happens with $t = 1, \ldots, T$ incremental tasks. For each incremental state $t$, we observe a training dataset $\mathcal{D}_t$ that consists of pairs $(\boldsymbol{x}_t, \boldsymbol{y}_t)$, in which $\boldsymbol{x}_t \in \mathcal{X}$ denotes an input image of size $N$, and $\boldsymbol{y}_t \in \mathcal{Y}_t^N$ denotes the corresponding ground-truth (GT) *pixel* labels. The label space $\mathcal{Y}_t = \{c_b\} \cup \mathcal{C}_t$ consists of the current classes in task $t$, $\mathcal{C}_t$, and the dummy background class $c_b$, that is assigned to all pixels that do not belong to $\mathcal{C}_t$. Thus, the $c_b$ label can be assigned to the objects with the *past* classes $\mathcal{C}^{1:t-1}$, the objects with the *future* classes $\mathcal{C}^{t+1:T}$, or the true background pixels.

After learning task $t$, the semantic segmentation model $f_{\boldsymbol{\theta}}^t$ is required to predict whether a pixel belongs to *all* the classes learned so far, $\mathcal{C}^{1:t} = \mathcal{C}^{1:t-1} \cup \mathcal{C}_t$, or the true background. As in other work, we assume the classes in each $\mathcal{C}_t$ are disjoint. Typically, the model is defined to be a mapping $f_{\boldsymbol{\theta}}^t : \mathcal{X} \to \mathbb{R}^{N \times |\mathcal{Y}^t|}$, in which $\mathcal{Y}^t = \mathcal{C}_d \cup \mathcal{C}^{1:t}$ with $\mathcal{C}_d$ containing dummy labels. All previous work [3, 7] simply set $\mathcal{C}_d = \{c_b\}$, but in our work, we also add the separate "*Unknown*" class label, $c_u$, to $\mathcal{C}_d$, hence, set $\mathcal{C}_d = \{c_b, c_u\}$. Later, we show defining this additional label $c_u$ in our model output plays a critical role in improving the learning capability for future classes. The architecture of $f_{\boldsymbol{\theta}}^t$ is typically a fully-convolutional network, which consists of a convolutional feature extractor, $h_{\boldsymbol{\psi}}^t$, followed by the final $1 \times 1$ classifier filters, $\{\phi_c^t\}_{c \in \mathcal{Y}^t}$, one for each output class in $\mathcal{Y}^t$.

The learning of $f_{\boldsymbol{\theta}}^t$ is done in conjunction with the previous model $f_{\boldsymbol{\theta}}^{t-1} : \mathcal{X} \to \mathbb{R}^{N \times |\mathcal{Y}^{t-1}|}$ to prevent forgetting during incrementally updating the model. Determining which output classifier (*e.g.*, softmax or sigmoid) to use for each pixel as well as how to transfer the knowledge of $f_{\boldsymbol{\theta}}^{t-1}$ to $f_{\boldsymbol{\theta}}^t$ (*e.g.*, knowledge distillation or model freezing) are design choices, and we elaborate our choices more in details in the subsequent sections. Furthermore, we denote $\mathcal{M}$ as the *exemplar memory*, which can store a small number of samples from past training data, $\mathcal{D}^{1:t-1}$, and use it for learning $f_{\boldsymbol{\theta}}^t$.

Once the learning of $f_{\boldsymbol{\theta}}^t$ is done, the prediction for pixel $i$ of an input image $\boldsymbol{x}$ is obtained by

$$\hat{\boldsymbol{y}}_i^t = \arg\max_{c \in \mathcal{Y}^t} f_{\boldsymbol{\theta}, ic}^t(\boldsymbol{x}),$$

and the performance is measured by the mean intersection-over-union (mIoU) metric for the classes in $\mathcal{Y}^t$. (Only in the evaluation phase, we merge $c_b$ and $c_u$ for computing mIoU of the BG class.)

### 3.2 Two Additional Challenges of CISS

In addition to the typical reason for causing catastrophic forgetting in CIL for classification, *i.e.*, the prediction bias due to the data imbalance, we note there are two additional unique challenges in CISS.

The first challenge, as mentioned in the Introduction and in previous work [3, 7], comes from the semantic drift of the background (BG) label. Namely, the ground-truth label associated with the pixels of the object can change depending on the incremental state. For example, when a pixel is labeled as "BG" at state $t$, it is possible that the corresponding pixel would be labeled as " sofa" at state $t - 1$ (*i.e.,* past class) or labeled as "dog" at state $t + 1$ (*i.e.*, future class), depending on the true object to which the pixel belongs. Therefore, naively learning with the "BG" target label for the pixel at state $t$ could cause either severe forgetting of the past class (*i.e.*, hurting stability) or interfering the learning of the future class (*i.e.*, hurting plasticity).

The second challenge stems from the fact that semantic segmentation is a multi-label prediction problem. Namely, for a given image, the segmentation model needs to output a *set* of classes, in contrast to the classification model which outputs only a single class. Therefore, the *precision* of the prediction for each pixel becomes important in addition to the recall as is reflected in the mIoU performance metric; *i.e.*, not only predicting a correct class for a pixel is important, but also not predicting a wrong class is important for the overall metric. That is, if the prediction for every pixel

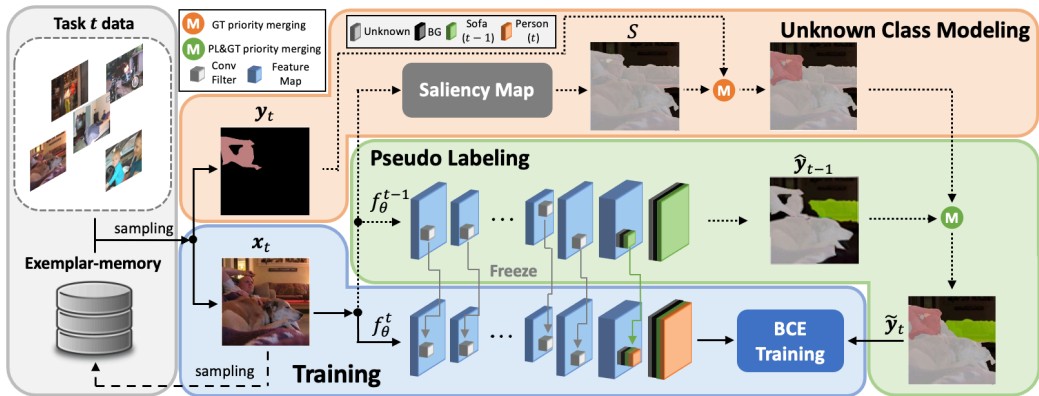

Figure 1: Overall procedure of SSUL. Given $(\boldsymbol{x}_t, \boldsymbol{y}_t) \in \mathcal{D}_t$, the augmented label $\tilde{\boldsymbol{y}}_t$ is first obtained by the "Unknown" class modeling and pseudo-labeling. Then, using $\tilde{\boldsymbol{y}}_t$ as a target, we update $f_{\boldsymbol{\theta}}^t$ with model freezing and BCE loss. The exemplar-memory is also updated with class-balanced sampling.

gets biased toward the current classes in $\mathcal{C}_t$, the mIoU's for the past classes in $\mathcal{C}^{1:t-1}$ *as well as* the current classes would significantly drop jointly. This point is exactly why even the mIoU's for the newly learned classes are *very low* in [3, 7]. Note this is in a stark difference with the classification, in which the accuracy of the current classes would remain high even with the severe bias and forgetting, since it is a single-label prediction problem.

To address above unique challenges of CISS, we devise our SSUL-M by carefully combining several ideas, of which overall procedure is outlined in Figure 1. We now elaborate on our three main contributions in details: 1) **label augmentation** for BG class with *Unknown* class and pseudo-labels, 2) **stable score learning** with model freeze and sigmoid output, and 3) usage of **tiny exemplar memory** with class-balanced sampling.

### 3.3 Contribution 1: Label Augmentation for BG Class

Here, we describe how we generate an augmented target label $\tilde{\boldsymbol{y}}_t \in (\mathcal{Y}^t)^N = \{c_b, c_u, \mathcal{C}^{1:t}\}^N$ given a training sample $(\boldsymbol{x}_t, \boldsymbol{y}_t)$ in $\mathcal{D}_t$. Recall that $\boldsymbol{y}_t \in (\mathcal{Y}_t)^N = \{c_b, \mathcal{C}_t\}^N$.

**Unknown class modeling** In order to handle the case in which the potential future class objects are labeled as BG, we propose to use a novel unknown class label, $c_u$, that is defined to designate any foreground objects that are not the current classes in $\mathcal{C}_t$. Specifically, as depicted in the top part of Figure 1, we first apply an off-the-shelf salient object detector [11] $S : \mathcal{X} \rightarrow \{0, 1\}^N$ to the input image $\boldsymbol{x}_t$, which assigns 1 to the pixels if they are salient (*i.e.*, part of a foreground object) and 0 otherwise. Then, we assign $c_u$ to the pixels that are BG-labeled but salient; namely, we set $\tilde{\boldsymbol{y}}_{t,i} = c_u$ if $(\boldsymbol{y}_{t,i} = c_b) \wedge (S(\boldsymbol{x}_t)_i = 1)$ for pixel $i$. In our experiments, we show this Unknown class label plays a critical role so that the base feature extractor can distinguish the representations of the potential future class objects and the true background, even before observing the class labels.

**Pseudo-labeling** Once augmenting with $c_u$ is done, we further augment with pseudo-labels generated from the previous model $f_{\boldsymbol{\theta}}^{t-1}$ to maintain the knowledge of the past classes, similarly as in [7] and as shown in the middle part of Figure 1. Namely, we denote $\hat{\boldsymbol{y}}_{t-1,i} = \arg\max_{c \in \mathcal{Y}^{t-1}} f_{\boldsymbol{\theta},ic}^{t-1}(\boldsymbol{x}_t)$ as the prediction of $f_{\boldsymbol{\theta}}^{t-1}$ for the $i$-th pixel and set $\tilde{\boldsymbol{y}}_{t,i} = \hat{\boldsymbol{y}}_{t-1,i}$ if

$$(\boldsymbol{y}_{t,i} = c_b) \wedge (\hat{\boldsymbol{y}}_{t-1,i} \in \mathcal{C}^{1:t-1}) \wedge (\boldsymbol{\mu}_i > \tau),$$

in which $\boldsymbol{\mu}_i = \max_{c \in \mathcal{C}^{1:t-1}} \sigma(f_{\boldsymbol{\theta},ic}^{t-1}(\boldsymbol{x}_t))$ stands for the confidence of prediction for $\hat{\boldsymbol{y}}_{t-1,i} \in \mathcal{C}^{1:t-1}$ where $\sigma(\cdot)$ is the sigmoid function, and $\tau$ is a threshold (set to 0.7). In words, we assign the pseudo-labels (*i.e.*, the predictions from the previous model) to the pixels that are BG-labeled only when the predictions are made to be the past object classes (excluding $c_b$ and $c_u$) with enough confidence.

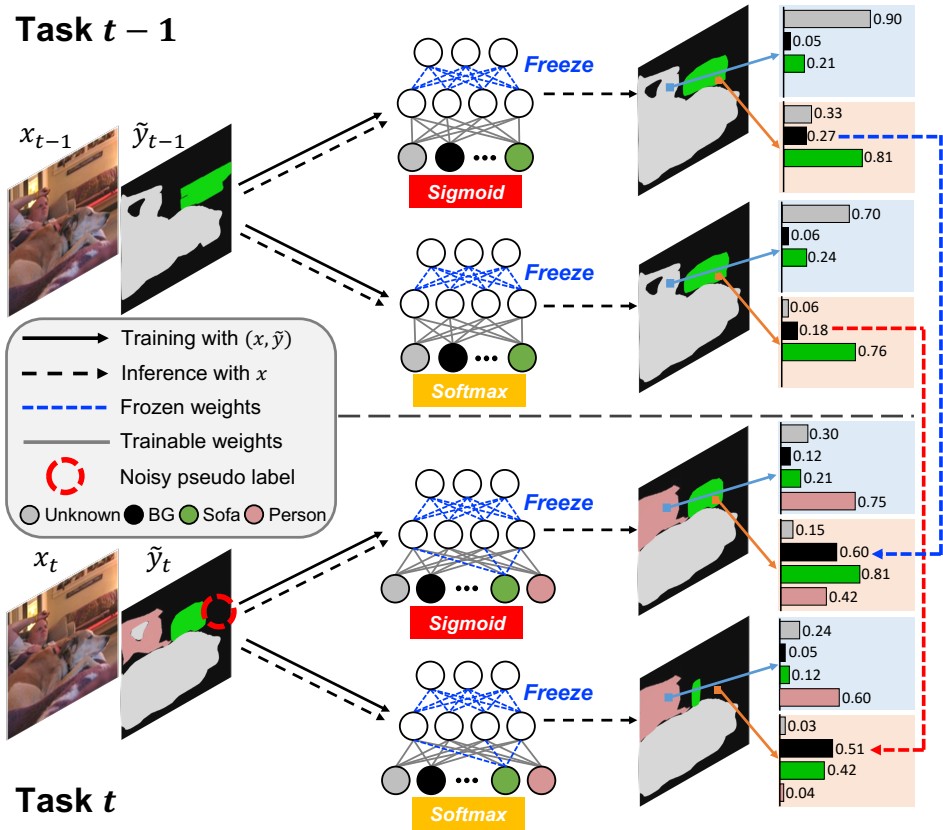

Figure 2: Comparison of the effect on the classification scores for sigmoid with binary cross-entropy (BCE) and softmax with cross-entropy (CE). For a pixel with noisy label $\tilde{y}_{t,i} = c_b$ (*i.e.,* BG label (black) when the true is "sofa" (green), see the red dashed circle at task $t$), even when the classifier for "sofa" ($\{\phi_c\}_{c\in\mathcal{C}^{1:t-1}}$) is frozen, the softmax would cause the BG class score to rise above the score for "sofa" (see the red dashed arrow), whereas the sigmoid would let the BG score rise only moderately (see the blue dashed arrow). Thus, the classification for the pixel could still become "sofa" for sigmoid, whereas softmax would make a false prediction as BG.

In summary, at the incremental state $t$, the augmented target label for the $i$-th pixel becomes

$$
\tilde{y}_{t,i} = \begin{cases}
y_{t,i} & \text{if } y_{t,i} \in \mathcal{C}_t \\
\hat{y}_{t-1,i} & \text{if } (y_{t,i} = c_b) \wedge (\hat{y}_{t-1,i} \in \mathcal{C}^{1:t-1}) \wedge (\mu_i > \tau) \\
c_u & \text{if } (y_{t,i} = c_b) \wedge (S(x_t)_i = 1) \wedge \{(\hat{y}_{t-1,i} \in \mathcal{C}_d) \vee (\mu_i \leq \tau)\} \\
c_b & \text{else,}
\end{cases}
\tag{1}
$$

in which the pseudo-label is generated only for the incremental state $t \geq 2$. Figure 1 shows a concrete example of augmentating $y_t$ to $\tilde{y}_t$, in which $\mathcal{C}_{t-1} = \{\texttt{sofa}\}$ and $\mathcal{C}_t = \{\texttt{person}\}$.

### 3.4 Contribution 2: Stable Score Learning

We argue that simply using $\tilde{y}_t$ as a target and training $f_\theta^t$ with softmax output layer would make the output scores too unstable as the incremental learning continues. The reason is because the augmented labels are *noisy* and the softmax computes the prediction probability in a *relative* way. As shown in our experiments, such instability significantly hurts the precision of the multi-label prediction. To that end, we propose the following three strategies for the stable learning of output scores.

**Model freezing** Instead of updating the full model with $\tilde{y}_t$ at every state $t$, we *freeze* the convolutional feature extractor, $h_\psi$, after initial learning ($t = 1$) as well as the classifiers for the past classes, $\{\phi_c\}_{c\in\mathcal{C}^{1:t-1}}$, and only learn the $1 \times 1$ filters $\phi_{c_b}^t$, $\phi_{c_u}^t$ and $\{\phi_c^t\}_{c\in\mathcal{C}_t}$. Such strict model freezing certainly can prevent forgetting, but is counter-intuitive from the plasticity viewpoint. However, thanks

to the unknown class $c_u$, it turns out the feature extractor $h_\psi$ can roughly learn the representations for the potential future class objects present with BG label in $\mathcal{D}_1$. Thus, it becomes sufficient to learn the decision boundaries for $c_b, c_u$, and $\mathcal{C}_t$ on those representations at state $t$.

**Sigmoid with binary cross entropy loss**  For learning $\phi_{c_b}^t$, $\phi_{c_u}^t$ and $\{\phi_c^t\}_{c \in \mathcal{C}_t}$, the choice of output layer and loss function becomes important to make the output score stable. A common choice is the softmax with cross-entropy loss, however, since the target labels for $c_b$ and $c_u$ in $\tilde{y}_t$ are noisy, we observe the relative scoring of softmax could cause significant forgetting of past classes $\mathcal{C}^{1:t-1}$. To see this, let $s_{ic} = [f_\theta^t(x_t)]_{ic}$ be the score for class $c$ at pixel $i$. When the softmax with cross-entropy (CE) is used, the gradient of the loss at $s_{ic}$ becomes $\partial \mathcal{L}_{\text{CE}}^t(\theta)/\partial s_{ic} = p_{ic} - \mathbb{1}\{c = \tilde{y}_{t,i}\}$, in which $p_{ic} = \exp(s_{ic})/(\sum_{c'} \exp(s_{ic'}))$. The issue occurs when $\tilde{y}_{t,i} = c_b$ or $c_u$, while the true label for the pixel should be the past classes in $\mathcal{C}^{1:t-1}$. (Thus, the pseudo-label $\hat{y}_{t-1,i}$ missed the pixel, which could often happen.) Then, the gradient descent learning will force $s_{ic_b}$ or $s_{ic_u}$ to become much higher than $\{s_{ic}\}_{c \in \mathcal{C}^{1:t-1}}$, the scores obtained from the *frozen* classifiers for past classes. Thus, at test time, for a similarly confusing pixel, the model would tend to predict as $c_b$ or $c_u$, hence, causing the forgetting of past classes even though their classifiers are fixed.

Therefore, we instead use the sigmoid output with binary cross-entropy (BCE) loss independently for each class. In that case, the gradient of the loss at $s_{ic}$ becomes $\partial \mathcal{L}_{\text{BCE}}^t(\theta)/\partial s_{ic} = \sigma(s_{ic}) - \mathbb{1}\{c = \tilde{y}_{t,i}\}$, hence, even for above noisy target label case, the scores $s_{ic_b}$ or $s_{ic_u}$ will only increase to a certain absolute level regardless of other class score values, $\{s_{ic}\}_{c \in \mathcal{C}^{1:t-1}}$. Thus, at test time, a similar pixel still may be predicted as a past class in $\mathcal{C}^{t-1}$ since the frozen past classifiers would still output a considerable score — this subtle difference between the loss functions is illustrated in Figure 2.

**Weight transfer from unknown class classifier**  Finally, for learning $\{\phi_c^t\}_{c \in \mathcal{C}_t}$, we initialize all the filters with $\phi_{c_u}^{t-1}$, the classifier learned for the unknown class $c_u$ at the previous state. The reasoning is that the classes in $\mathcal{C}_t$ would have been labeled as $c_u$ (as the potential future classes) in state $t-1$, hence, such weight transfer from $\phi_{c_u}^{t-1}$ would yield stable and fast learning of $\{\phi_c^t\}_{c \in \mathcal{C}_t}$.

### 3.5  Contribution 3: Tiny Exemplar Memory

Using exemplar-memory to store a small portion of past training data for CIL is backed with a theoretical finding [19] as well as strong empirical results [27, 30, 2, 1, 4] for classification.

Hence, we propose to use it for CISS as well with a tailored *class-balanced* sampling strategy. The main rationale of using the memory is to make sure to include at least one sample with correct GT label per each class in $\mathcal{C}^{1:t-1}$ in the training set for state $t$.

Namely, even though the pseudo-label $\hat{y}_{t-1}$ can provide labels for the classes in $\mathcal{C}^{1:t-1}$, it is also possible that the given image $x_t$ would never contain object cues for $\mathcal{C}^{1:t-1}$. In such a case, even with model freezing and stable score learning, when the confidence for the new class is learned to be high for a pixel (potentially for an old class), the prediction for the pixel could get biased toward the new class, causing the forgetting of the old class. Therefore, by denoting $M = |\mathcal{M}|$, after learning incremental state $t-1$, we sample $M/|\mathcal{C}^{1:t-1}|$ data points from $\mathcal{D}_{t-1}$, and store them in the memory $\mathcal{M}$ by removing equal number of samples per each class in $\mathcal{C}^{1:t-2}$ from $\mathcal{M}$. In this way, $\mathcal{M}$ always contains at least one sample from each class in $\mathcal{C}^{1:t-1}$, and we show in the experiments that this class-balanced strategy is more helpful than random sampling [4] as is done for CIL for classification.

---

**Algorithm 1:** Pseudo code of SSUL-M

1: **Require** $f_\theta$, $\mathcal{M}$, $T$, $n_{\text{epochs}}$, batch size $K$
2: $f_\theta \leftarrow$ Initialize $h_\psi$, $\phi_{c_b}$, $\phi_{c_u}$
3: Initialize $\mathcal{M}$
4: **for** $t \leftarrow 1, ..., T$ **do**
5:     $f_\theta^t \leftarrow$ Initialize $\{\phi_c^t\}_{c \in \mathcal{C}_t}$
6:     **if** $t \neq 1$ **then**
7:         Freeze $h_\psi$, $\{\phi_c^t\}_{c \in \mathcal{C}^{1:t-1}}$
8:         $\{\phi_c^t\}_{c \in \mathcal{C}_t} \leftarrow$ `Weight Transfer`($\phi_{c_u}^{t-1}$)
9:     **end if**
10:    **for** $n \leftarrow 1, ..., n_{\text{epochs}}$ **do**
11:       **for** $B_n \overset{K/2}{\sim} \mathcal{D}_t$ **do**
12:          $B_\mathcal{M} \overset{K/2}{\sim} \mathcal{M}$
13:          $\tilde{B}_n, \tilde{B}_\mathcal{M} \leftarrow$ `Label.Aug.`($B_n, B_\mathcal{M}$)
14:          $\theta \leftarrow$ SGD($\tilde{B}_n \cup \tilde{B}_\mathcal{M}, \mathcal{L}_{\text{BCE}}^t(\theta)$)
15:       **end for**
16:    **end for**
17:    Update $\mathcal{M}$
18: **end for**
19: **return** $f_\theta^t$

---

### 3.6 Summary

We summarize our SSUL-M algorithm in Algorithm 1, in which our contributions described in Section 3.3∼ Section 3.5 are denoted in the typewriter font. Namely, `Label.Aug.` stands for generating $\tilde{y}_t$ as in (1) for the selected batches $\mathcal{B}_n$ and $\mathcal{B}_{\mathcal{M}}$, "`Freeze`, $\mathcal{L}^t_{\text{BCE}}$, and `Weight Transfer`" denote the methods for the stable score learning described in Section 3.4, and `Update` $\mathcal{M}$ denote the exemplar-memory maintenance with the class balanced sampling as mentioned in Section 3.5. Note we are constructing a mini-batch by sampling equal amount of data from $\mathcal{D}_t$ and $\mathcal{M}$, hence, the samples in $\mathcal{M}$ act as anchor points to improve the precision of the predictions.

## 4 Experiments

### 4.1 Experimental Setting

**Dataset**  We followed the experimental setting of [3] and evaluated our method using Pascal-VOC 2012 [9] and ADE20K [32] datasets. Originally, [3] set two experimental protocols, *disjoint* and *overlapped*, but we believe the latter is more realistic and challenging. Therefore, we evaluated on the *overlapped* setup only. **Pascal VOC 2012** contains 20 foreground object classes and one background class, and **ADE20K** consists of 150 classes of both stuff and objects. We consider several incremental learning scenarios for each dataset, from the scenarios considered by the other baselines to newly proposed *challenging* scenarios with larger number of incremental states. A more detailed description on the datasets is introduced in the Supplementary Material (S.M.).

**Evaluation Metrics**  We use the mean Intersection-over-Union (mIoU) as our evaluation metric, which computes the IoU for each class then computes the average over the classes. The IoU is defined as $IoU = \textit{true-positive}/(\textit{true-positive} + \textit{false-positive} + \textit{false-negative})$.

**Implementation Details**  For all experiments, following other works [7, 3], we use a Deeplab-v3 segmentation network [5] with a ResNet-101 [10] backbone pre-trained on ImageNet [6]. We optimize the network using SGD with an initial learning rate of $10^{-2}$ and a momentum value of 0.9 for all CISS steps. Also, we set the learning rate schedule, data augmentation, and output stride of 16 following [5] for all experiments. For each incremental state $t$, we train the network for 50 epochs for Pascal VOC with a batch-size of 32 and 60 epochs for ADE20K with a batch-size of 24. We tune the hyperparameters using 20% of the training set as a validation set and report the final results on the standard test set. For the exemplar memory, we utilized memory with a fixed size of $|\mathcal{M}| = 100$ for Pascal VOC and $|\mathcal{M}| = 300$ for ADE20K. To highlight the effect of the exemplar-memory, we report the results of the two versions of our method — SSUL (without memory) and SSUL-M (with memory). For the saliency-map detector to generate the Unknown class label, we employed DSS [11] pretrained on MSRA-B dataset [22], which contains 5,000 labels for salient objects. The experiments were implemented in PyTorch [26] 1.7 with CUDA 10.1 and CuDNN 7 using two NVIDIA V100 GPUs, and all experiments were conducted with NSML [15] framework. More information on the experimental details are in the S.M.

**Baselines**  For a representative of the general regularization-based continual learning method, we select EWC [18] and LWC [20]-MC and applied them to each experimental setup of CISS. For CISS specific baselines, we compared with ILT [24], MiB [3] and PLOP [7], and the Joint Training (Joint) result is also given as an upper bound. Note that PLOP [7] is the current state-of-the-art. We reproduced the results of all baselines using the official code provided by the authors of [7].

### 4.2 Experimental results on benchmark dataset

**Pascal VOC 2012**  Following [7, 3], we evaluate our method on four different scenarios, *i.e.*, {10-1, 15-1, 15-5, and 19-1} as well as a more challenging scenario, {5-3}. The numbers in each scenario denote the number of classes to be trained for each state. For example, VOC 5-3 means learning 5 classes at the base task ($t = 1$), and then incrementally learning 3 classes five times (hence, $T = 6$).

In Table 1, we observe our SSUL consistently outperforms the baselines with *huge* margin in all scenarios, even without using the exemplar-memory. Furthermore, the gap widens particularly for more challenging scenarios, namely, for the cases in which the base task has fewer classes and the number of tasks is larger. Note although MiB [3] and PLOP [7] show robustness for simple 2 tasks scenarios (19-1 and 15-5), their performance are *rapidly* dropped in more challenging and practical

Table 1: Experimental results on Pascal VOC 2012.

| Method | VOC 10-1 (11 tasks) | | | VOC 15-1 (6 tasks) | | | VOC 5-3 (6 tasks) | | | VOC 19-1 (2 tasks) | | | VOC 15-5 (2 tasks) | | |
| --- | --- | --- | --- | --- | --- | --- | --- | --- | --- | --- | --- | --- | --- | --- | --- |
| | 0-10 | 11-20 | all | 0-15 | 16-20 | all | 0-5 | 6-20 | all | 0-19 | 20 | all | 0-15 | 16-20 | all |
| EWC [18] | - | - | - | 0.30 | 4.30 | 1.30 | - | - | - | 26.90 | 14.00 | 26.30 | 24.30 | 35.50 | 27.10 |
| LwF-MC [20] | 4.65 | 5.90 | 4.95 | 6.40 | 8.40 | 6.90 | 20.91 | 36.67 | 24.66 | 64.40 | 13.30 | 61.90 | 58.10 | 35.00 | 52.30 |
| ILT [24] | 7.15 | 3.67 | 5.50 | 8.75 | 7.99 | 8.56 | 22.51 | 31.66 | 29.04 | 67.75 | 10.88 | 65.05 | 67.08 | 39.23 | 60.45 |
| MiB [3] | 12.25 | 13.09 | 12.65 | 34.22 | 13.50 | 29.29 | 57.10 | 42.56 | 46.71 | 71.43 | 23.59 | 69.15 | 76.37 | 49.97 | 70.08 |
| PLOP [7] | 44.03 | 15.51 | 30.45 | 65.12 | 21.11 | 54.64 | 17.48 | 19.16 | 18.68 | 75.35 | 37.35 | 73.54 | 75.73 | 51.71 | 70.09 |
| SSUL | 71.31 | 45.98 | 59.25 | 77.31 | 36.59 | 67.61 | 72.44 | 50.67 | 56.89 | 77.73 | 29.68 | 75.44 | 77.82 | 50.10 | 71.22 |
| SSUL-M | 74.02 | 53.23 | 64.12 | 78.36 | 49.01 | 71.37 | 71.27 | 53.21 | 58.37 | 77.83 | 49.76 | 76.49 | 78.40 | 55.80 | 73.02 |
| Joint | 78.41 | 76.35 | 77.43 | 79.77 | 72.35 | 77.43 | 76.91 | 77.63 | 77.43 | 77.51 | 77.04 | 77.43 | 79.77 | 72.35 | 77.43 |

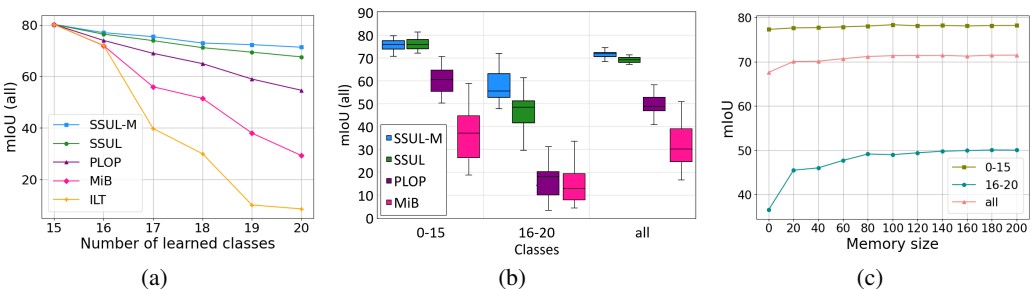

Figure 3: (a): mIoU evaluation on VOC 15-1, (b): mIoU distributions for 20 different class-orderings for VOC 15-1, (c): mIOU on VOC 15-1 with varying memory size $|\mathcal{M}|$.

scenarios (10-1, 5-3, 15-1). More specifically, Figure 3(a) shows the mIoU evolution for 15-1 scenario at each incremental step, and the baselines suffers from a significant drop of mIoU, as the new classes are incrementally learned. In contrast, SSUL improves mIoUs for both base (0-15) and newly learned (16-20) classes significantly, showing much slower drop of mIoU. These results show that as long as the label augmentation of BG class with Unknown class and pseudo-labels is properly done, our stable score learning is very effective for CISS. Particularly, we observe that model freeze, which is believed to be not effective in CIL, is much more effective than the KD used in other baselines.

Furthermore, we observe our SSUL-M, which uses exemplar-memory, further strengthens SSUL significantly, particularly for the newly learned classes (*i.e.*, for $t \geq 2$). For example, in VOC 15-1, by storing only $5 \sim 6$ images per class in $\mathcal{M}$, the mIoU for classes 16-20 improved about 12%. This confirms our intuition that the samples in the memory act as "anchors" to improve the precision of the predictions, hence, prevent forgetting. We can also clearly observe this improvement in Figure 4. As an additional experiment, we also conducted the experiment on the case of further reducing the number of base classes, such as 5-1 and 2-1, and we again observe that our SSUL and SSUL-M surpass other baselines. A detailed result on this additional experiment can be found also in the S.M.

To verify the robustness of each method on various class orderings, we experimented on the 20 difference orderings on VOC 15-1 scenario, as proposed in [7]. Figure 3(b) plots the mIoU distributions for different methods, and we clearly observe both SSUL and SULL-M not only achieve higher mIoUs, but also have smaller variations compared to the baselines. In addition, we note SULL achieves $\times 1.5$ faster training time compared to PLOP due to network freezing.

**Qualitative analysis** In Figure 4, we visualized the qualitative results for four images from VOC 15-1 task. We observe that PLOP partly maintains knowledge learned from the base task (*bird* and *person*), however, it becomes fatal in forgetting the classes learned in the intermediate steps, such as plant and sheep. In addition, PLOP frequently produces many false-positive predictions, lowering the mIoU of several classes. (E.g., see *bird* in Step 5.) On the other hand, we observe SSUL and SSUL-M both maintains the previously learned classes with high stability and effectively learns new classes. For example, the *sheep* class is accurately learned at Step 3 and not forgotten afterwards. Moreover, we observe SULL-M achieved further improvement of both plasticity and stability over SULL, especially by reducing the false-positive predictions.

**ADE20K** Unlike VOC 2012, ADE20K is densely labeled for both stuff and thing with 150 classes. It means that the class definition in ADE20K is clearer therefore, it naturally make reduce the concern about the semantic drift of BG label. To make efficient use of this prior knowledge of dataset, we

Table 2: Experimental results on ADE20K.

| Method | ADE 100-5 (11 tasks) | | | ADE 100-10 (6 tasks) | | | ADE 100-50 (2 tasks) | | | ADE 50-50 (3 tasks) | | |
|---|---|---|---|---|---|---|---|---|---|---|---|---|
| | 0-100 | 101-150 | all | 0-100 | 101-150 | all | 0-100 | 101-150 | all | 0-50 | 511-150 | all |
| ILT [24] | 0.08 | 1.31 | 0.49 | 0.11 | 3.06 | 1.09 | 18.29 | 14.40 | 17.00 | 3.53 | 12.85 | 9.70 |
| MiB [3] | 36.01 | 5.66 | 25.96 | 38.21 | 11.12 | 29.24 | 40.52 | 17.17 | 32.79 | 45.57 | **21.01** | 29.31 |
| PLOP [7] | 39.11 | 7.81 | 28.75 | **40.48** | 13.61 | 31.59 | **41.87** | 14.89 | 32.94 | 48.83 | 20.99 | **30.40** |
| SSUL | 39.94 | **17.40** | 32.48 | 40.20 | **18.75** | 33.10 | 41.28 | **18.02** | 33.58 | 48.38 | 20.15 | 29.56 |
| SSUL-M | **42.86** | 17.78 | **34.56** | **42.86** | 17.66 | **34.46** | 42.79 | 17.54 | **34.37** | **49.12** | 20.10 | 29.77 |
| Joint | 44.30 | 28.20 | 38.90 | 44.30 | 28.20 | 38.90 | 44.30 | 28.20 | 38.90 | 51.10 | 33.30 | 38.90 |

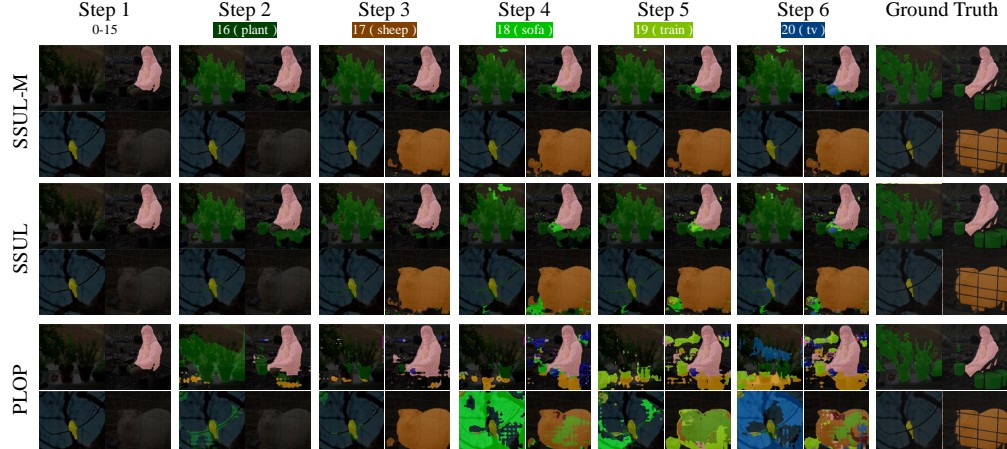

Figure 4: Qualitative results of SSUL-M, SSUL and PLOP on VOC 15-1 scenario.

consider unlabeled pixels as unknown class without using the saliency detector and enlarge the size of memory to $|\mathcal{M}| = 300$. We evaluated our method in four different scenarios, *i.e.*, {100-5, 100-10, 50-50, 100-50} as in [7, 3]. Table 2 again shows that SSUL achieves superior performance in the more challenging tasks (100-5, 100-10) than other methods. We believe that, this result demonstrates SSUL is also quite effective in CISS for a densely labeled dataset, without any extra saliency detector. The qualitative analysis on ADE20K is provided in the S.M.

## 4.3 Ablation Study

**Ablation study on proposed components of SSUL** Here, we analyze the effect of each proposed component of SSUL on VOC 15-1 scenario. Table 3 compares the mIoU of each ablation case, without memory, and the first low shows the result of SSUL with full components. Firstly, when the Unknown class is removed, we clearly observe that the mIoU's of both the first task (0-15) and subsequent classes (16-20) decrease. Hence, this clearly demonstrates the advantage of using the Unknown class label, in terms of increasing both the plasticity and stability. Secondly, note that model freezing has a significant impact on the performance of CISS. Specifically, it not only prevents catastrophic forgetting on the first task, but also plays a critical role to learn new classes well. Finally, we observe when Softmax+CE instead of Sigmoid+BCE is used (the last ablation), the forgetting of the first task (0-15) drops more significantly despite model freezing. We believe this confirms our intuition in Section 3.4 on why Sigmoid+BCE can lead to more stable score learning.

**Saliency-map detector and weight transfer** The first two columns in Table 4 shows the ablation study on saliency-map detector for Unknown class and weight transfer. We compare the result of a default neural network-based saliency map extractor, DSS [11] with ground-truth and a random forest-based DRFI [13]. We observe the differences of the mIoU's among them are quite small, therefore, we believe that the quality of saliency map is not a significant factor of our method. The ablation study on weight transfer demonstrates weights for new classes initialized with $\phi_{c_u}^{t-1}$ is most effective to learn it. This result may at first look counter-intuitive since learning the final linear layer $\phi_c^t$ should not be sensitive on the initialization. However, given the *noisy* pseudo-label for learning, a quicker convergence of $\phi_c^t$ using the warm-start weight $\phi_{c_u}^{t-1}$ would effectively prevent the forgetting of past class, which could be caused by fitting the noisy pseudo-label for many epochs. More details on the effect of weight transfer is proposed in S.M.

Table 3: Ablation study for SSUL on VOC 15-1 about PL (pseudo labeling), Unknown (unknown class modeling & weight transfer), Freeze (model freezing), and Sigmoid+BCE (stable score learning).

| | | | | 15-1 (6 tasks) | | |
|---|---|---|---|---|---|---|
| PL | Unknown | Freeze | Sigmoid+BCE | 0-15 | 16-20 | all |
| ✓ | ✓ | ✓ | ✓ | **77.31** | **36.59** | **67.61** |
| ✓ | ✗ | ✓ | ✓ | 73.42 | 21.79 | 61.12 |
| ✓ | ✓ | ✗ | ✓ | 53.56 | 14.48 | 44.25 |
| ✓ | ✓ | ✓ | ✗ | 61.42 | 22.97 | 52.26 |

Table 4: Ablation study for saliency-map detector (left), weight transfer (mid), and memory sampling (right) on VOC 15-1.

| Saliency-map detector | | | | Weight transfer | | | | Memory sampling | | | |
|---|---|---|---|---|---|---|---|---|---|---|---|
| Methods | 0-15 | 16-20 | all | Methods | 0-15 | 16-20 | all | Methods | 0-15 | 16-20 | all |
| DRFI [13] | 76.46 | 32.53 | 66.00 | $random \rightarrow \phi_c^t$ | 73.73 | 23.99 | 61.89 | random | 78.61 | 38.87 | 69.15 |
| DSS [11] | 77.31 | 36.59 | 67.61 | $\phi_{c_b}^{t-1} \rightarrow \phi_c^t$ | 73.29 | 23.70 | 61.48 | class-balanced | **78.36** | **49.01** | **71.37** |
| ground-truth | 78.42 | 40.41 | 69.36 | $\phi_{c_u}^{t-1} \rightarrow \phi_c^t$ | **77.31** | **36.59** | **67.61** | - | - | - | - |

**Memory size and sampling rule**   The right column row of Table 4 shows the result of the mIoU on two sampling rules. We observe that, compared to random sampling, our proposed class-balanced sampling achieves better mIoU, particularly for the newly learned classes. We believe the reason is class-balanced sampling ensures at least one sample per class, therefore, it prevents the forgetting of minority classes than random sampling, which can miss certain classes in $\mathcal{M}$. Finally, Figure 3(c) shows the dependency of SSUL-M on the memory size. It illustrates that tiny exemplar-memory for CISS significantly helps to increase the mIoU for the newly learned classes (16-20) than the base classes (0-15). Moreover, we observe that after sufficiently large $|\mathcal{M}|$ the mIoU performance becomes robust.

## 5   Concluding Remarks and Limitation

We proposed a new class-incremental learning method SSUL-M (Semantic Segmentation with Unknown Label with Memory) for semantic segmentation. In order to address two additional challenges of CISS, we made three main contributions — label augmentation with Unknown class labels, stable score learning, and tiny exemplar memory. They all were convincingly shown to be very effective in various CISS scenarios and our SSUL-M significantly outperformed baselines.

While promising, we admit our work also has some limitations as follows. First, our unknown class modeling may not be satisfactory for some "stuff" segmentation tasks since saliency-maps are mainly targeted for "things" (or objects). Second, our model freezing may harm plasticity and cause the model suffer from learning new classes especially when they are not captured by the unknown class label during the base task. Together with our insights from the analyses of SULL-M, we believe attempting to address above limitations would lead to a fruitful future research directions for the CISS problem.

## Acknowledgement

This work was done while Sungmin Cha did a research internship at NAVER AI Lab. This work was also supported in part by the New Faculty Startup Fund from Seoul National University, NRF Mid-Career Research Program [NRF-2021R1A2C2007884], IITP grant [NO.2021-0-01343, Artificial Intelligence Graduate School Program (Seoul National University)] funded by the Korea government(MSIT), and and SNU-NAVER Hyperscale AI Center. The authors thank NAVER Smart Machine Learning (NSML) team for the GPU support. Taesup Moon also thanks the support from Automation and Systems Research Institute (ASRI) at Seoul National University.

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
