# Supplementary Materials for
# SSUL: Semantic Segmentation with Unknown Label for Exemplar-based Class-Incremental Learning

**Sungmin Cha**[1,2]*, **Beomyoung Kim**[3]*, **Youngjoon Yoo**[2,3], **and Taesup Moon**[1]
[1] Department of Electrical and Computer Engineering, Seoul National University
[2] NAVER AI Lab, [3] Face, NAVER Clova
sungmin.cha@snu.ac.kr, {beomyoung.kim, youngjoon.yoo}@navercorp.com,
tsmoon@snu.ac.kr

## 1 Additional Discussions

### 1.1 On model freezing and plasticity

The model freezing may seem too restrictive as it may focus too much on maintaining the performance of the old classes and limit the capability of learning new classes. However, in our ablation study (Table 3 in the manuscript), by comparing 1st and 3rd line of the table, we observe that model freezing significantly improves the mIoU not only for the old classes, but also for the new classes. This somewhat counterintuitive phenomenon mainly has to do with the second challenge mentioned above — $i.e$, since the predictions for the old classes remain accurate, the mIoU for the new classes also improves by limiting the false positives.

With such a positive effect of model freezing on the mIoU for the new classes, the unknown class label is introduced to compensate for the restrictiveness of complete freezing of the feature extractor in learning new classes and deal with the BG shift issue. Comparing the 1st and 2nd line of Table 3 in the manscript shows that the unknown class clearly further improves the performance for the new classes by effectively learning the representations of potential future classes (embedded in the unknown class). Note the result of SSUL (for VOC 15-1 task) is better than that of PLOP not only for the old classes, but also for the new classes when the unknown class label is used.

Finally, the learning capability for the new classes significantly gets boosted with the usage of the exemplar-memory. That is, for the VOC 15-1 task in Table 1, the mIoU for the new classes improves from 36.59% (SSUL) to 49.01% (SSUL-M). This again is due to the fact that the ground-truth labels for the old classes in the memory help make less false positive errors for the new classes.

In summary, although the model freezing may seem somewhat counterintuitive for achieving the plasticity, we convincingly show that it is very effective not only for maintaining the performance of the old classes, but also for learning the new classes in the CISS problem.

### 1.2 The case for stuff (area-like) and thing (object-like) classes in CISS

One of our main target issues is the semantic drift between true background and future class, and we alleviate the issue using the unknown class modeling with saliency maps. We believe that the applicability of the saliency maps depends on the dataset condition, that is, the matter is how many true backgrounds are included in the dataset.

Firstly, the segmentation datasets including only the object/thing (e.g., Pascal VOC) contain a huge number of true backgrounds. In this case, we showed that our unknown modeling with saliency maps

---

*Equal contribution.

35th Conference on Neural Information Processing Systems (NeurIPS 2021).

is greatly helpful to alleviate the semantic drift between true background and future class (in Table 1, Figure 3 and Figure 4 of the manuscript). Secondly, if the segmentation dataset including both the thing and stuff contains few true background pixels (e.g., ADE20K), the semantic drift rarely occurs. In this case, we can apply our unknown modeling without saliency maps and achieve a reasonable performance as in Table 2 of the manuscript. Finally, if the segmentation dataset including the thing and stuff contains many true backgrounds, the saliency maps may not be helpful.

Nevertheless, we respectfully argue that the final case, that is, the dataset supports stuff segmentation yet contains many true backgrounds is not a common condition. Most famous datasets (e.g., Cityscape [4], COCO-stuff [2], Pascal Context [10], and NYU [11]) contain few true backgrounds, therefore, our unknown modeling can widely be utilized in the same way as applied to ADE20K.

### 1.3 The details on sampling strategy for tiny examplar memory

Here, we describe the details of examplar memory sampling strategy. We denoted that the examplar memory as $\mathcal{M}$ and the size of examplar memory as $M = |\mathcal{M}|$. After learning the incremental step $t - 1$ ($t > 0$), we sample the $\mathcal{M}$ for $\mathcal{C}^{1:t-1}$ in $\mathcal{D}_{t-1}$. For the *random* sampling strategy, we randomly select the data from $\mathcal{D}_{t-1}$ until the size of $\mathcal{M}$ becomes $M$. However, as mentioned in the ablation study section of the main paper, majority classes (e.g., person, dog, car) are mainly sampled in $\mathcal{M}$, and none of the minority classes (e.g., TV, plant) may be sampled in $\mathcal{M}$, inducing the catastrophic forgetting of the minority classes. To alleviate the problem, we designed the *class-balanced* sampling strategy; for each class in $\mathcal{C}^{1:t-1}$, we collect $M/|\mathcal{C}^{1:t-1}|$ samples in $\mathcal{D}_{t-1}$ and store them in the $\mathcal{M}$. When the $|\mathcal{M}|$ is not equal to $M$, we randomly drop or supplement the samples in $\mathcal{M}$ until $|\mathcal{M}| = M$. In addition, at the $t + 1$ step, we remove some samples of each class (i.e., $\mathcal{C}^{1:t-1}$) in $\mathcal{M}$ and add samples for $\mathcal{C}_t$ to $\mathcal{M}$, so that $\mathcal{M}$ contains $M/|\mathcal{C}^{1:t}|$ samples per each class. This strategy ensures that $\mathcal{M}$ always contains at least one sample for each class, and we verified that the effectiveness of the *class-balanced* sampling strategy in the Table 4 of the manuscript.

## 2    Additional Details of Datasets

**Pascal VOC 2012** consists of 13,487 images, and it is divided as 10,582 images for training, 1,449 images for validation and 1,456 images for test dataset. **ADE20K** is a large scale dataset for semantic segmentation of scenes, including 25,210 images. It is also grouped as 20,210 images for the training set, 2,000 images for the validation set, and 3,000 images for the testing set. As stated in the manuscript, we followed exactly same experimental settings with PLOP [5].

## 3    The More Details of Experiments on Pascal VOC 2012

### 3.1    The experimental result on reducing the initial number of classes

The result for the case in which the initial class set is small can be found in Table 1 (VOC 5-3 task) in the manuscript, and we observe our method still significantly surpasses the baselines. Furthermore, we carried out additional experiments with small initial classes (i.e., VOC 5-1, 2-1, and 2-2 tasks), as shown in Table 1. We again observe that our SSUL consistently outperforms other baselines by a large margin even for such extreme cases, which confirms the effectiveness of our SSUL method.

Table 1: The experiment result on the small initial number of classes.

| Method | VOC 10-1 (11 tasks) | | | VOC 5-1 (16 tasks) | | | VOC 2-1 (19 tasks) | | | VOC 2-2 (10 tasks) | | |
|---|---|---|---|---|---|---|---|---|---|---|---|---|
| | 0-10 | 11-20 | all | 0-5 | 6-20 | all | 0-2 | 3-20 | all | 0-2 | 3-20 | all |
| MiB [3] | 12.25 | 13.09 | 12.65 | 11.47 | 9.45 | 10.03 | 21.57 | 7.93 | 9.88 | 41.11 | 23.35 | 25.89 |
| PLOP [5] | 44.03 | 15.51 | 30.45 | 0.12 | 9.00 | 6.46 | 0.01 | 5.22 | 4.47 | 24.05 | 11.92 | 13.66 |
| SSUL | **71.31** | **45.98** | **59.25** | **69.32** | 40.38 | 48.65 | **62.35** | 34.32 | 38.32 | **62.38** | 42.46 | 45.31 |
| SSUL-M | **74.02** | **53.23** | **64.12** | **71.86** | 48.41 | 55.11 | 60.49 | 42.11 | 44.74 | 58.85 | 45.82 | 47.68 |

## 3.2 The additional ablation study on weight transfer

One may think the initialization of the final linear layer would not matter since the feature extractor is frozen. However, as also observed in prior work on CIL [1, 6], the number of training iterations or the level of overfitting during learning seems to play an important role in finding the right trade-off between learning new classes and not forgetting the old classes. To that regard, we did additional experiments checking the effect of initialization and the number of training iterations in Table 2. Namely, we simply did the random initialization for $\phi_c^t$ and increased the number of iterations ($\times 1$, $\times 2$, $\times 4$) and compared with our weight transfer ($\times 1$) results. From the table, we observe that the random initialization cannot reach the performance of weight transfer even with a larger number of iterations. We argue this is a similar phenomenon as what we observe in ordinary CIL. Namely, for the new class sigmoid, it is true that the linear weight would converge to the same solution regardless of the initialization. However, the number of iterations to reach the solution would differ – if we do the weight transfer, it will converge quicker, whereas if randomly initialized, a larger epoch would be necessary. Now, in the latter case, running with a larger epoch would result in the overfitting of the weights for $c_u$ and $c_b$ to the noisy pseudo label. Then, it will cause overconfident predictions for the Unknown or Background classes, which would penalize the overall mIoU. On the other hand, when only a small number of iterations are used for the random initialization, the prediction for the new class would become inaccurate, which again would hurt the overall mIoU.

Table 2: The additional ablation study for weight transfer

| Head Init | Iterations | VOC 15-1 (6 tasks) | | |
| --- | --- | --- | --- | --- |
| | | 0-15 | 16-20 | all |
| Random | $\times 1$ | 73.73 | 23.99 | 61.89 |
| Random | $\times 2$ | 73.89 | 23.08 | 62.45 |
| Random | $\times 4$ | 72.22 | 22.38 | 61.14 |
| Weight Transfer | $\times 1$ | **77.31** | **36.59** | **67.61** |

## 3.3 The effect of using the saliency-detector

In order to observe the effect of using the saliency-detector on the performance of other baselines, we additionally experimented with a variant of PLOP. That is, we implemented PLOP with saliency map, which only used the pseudo-labels that exist in the salient regions, similarly as in ours. As we can see in Table 3, PLOP with saliency map slightly improves the original PLOP by 2.37% (*i.e.*, 54.64% → 57.01%). However, such result is still significantly lower than the result of SSUL (67.61%), underscoring the fact that our gain is not merely stemming from the additional data required for training the saliency detector.

Furthermore, even in our SSUL, we note that just naively using the saliency map would not automatically bring the mIoU gain. Namely, the saliency detector is closely used with our unknown class modeling, however, as we can see in Table 3 (2nd line) and Table 4 (weight transfer column) in the manuscript, if the correct weight transfer is not used, the mIoU becomes 61.89% even with the unknown class, which is not much higher than the result without the saliency detector and unknown class, *i.e.*, 61.12%. Note we can enjoy the improvement to 67.61% (+6.49%) only when the proper weight transfer is applied.

Table 3: The additional experiments using the saliency-detector.

| | Saliency | VOC 15-1 (6 tasks) | | |
| --- | --- | --- | --- | --- |
| | | 0-15 | 16-20 | all |
| PLOP | ✗ | 65.12 | 21.11 | 54.64 |
| PLOP | ✓ | 67.25 | 24.22 | 57.01 |
| SSUL | ✓ | **77.31** | **36.59** | **67.61** |

## 3.4 Additional hyperparameter search for $\tau$

When we generate the pseudo-label, we set the threshold $\tau$ on the output of previous model $\mu_i = \max_{c \in \mathcal{C}^{1:t-1}} \sigma\big(f_{\boldsymbol{\theta},ic}^{t-1}(\boldsymbol{x}_t)\big)$ for the confident pseudo-label. Here, we conduct an experiment to analyze

the effect of the hyperparameter $\tau$ in Table 4. When $\tau = 0$, which means without the thresholding, we can still achieve a competitive performance of 67.44% all mIoU, however, we found that some noisy labels are included in the pseudo-label. To increase the confidence of the pseudo-label and reduce the noisy labels, we set the $\tau$ and found that enough high $\tau$ of 0.7 can slightly boost the performance. In addition, as in Table 4, we found that the proposed method is robust to the $\tau$.

Table 4: Effect of threshold $\tau$.

| | VOC 15-1 (6 tasks) | | |
|---|---|---|---|
| $\tau$ | 0-15 | 16-20 | all |
| 0.0 | 77.31 | 35.85 | 67.44 |
| 0.3 | 77.51 | 36.06 | 67.64 |
| 0.5 | 77.34 | 35.96 | 67.49 |
| 0.7 | 77.28 | 37.51 | **67.81** |
| 0.9 | 77.31 | 36.59 | 67.61 |
| 1.0 | 75.08 | 40.08 | 66.74 |

## 3.5 The experimental result on disjoint setup

As mentioned in seminar works [3, 5], the disjoint setup has an additional constraint that only the classes observed so far are present in the input image, which is not necessarily practical. However, in this section, we provide an additional experiment result on the disjoint setup on Table 5. We clearly observe that SSUL and SSUL-M again achieve superior results compared to other baselines.

Table 5: Experimental results on Pascal VOC 2012 for *disjoint* setup.

| Method | VOC 10-1 (11 tasks) | | | VOC 15-1 (6 tasks) | | | VOC 19-1 (2 tasks) | | | VOC 15-5 (2 tasks) | | |
|---|---|---|---|---|---|---|---|---|---|---|---|---|
| | 0-10 | 11-20 | all | 0-15 | 16-20 | all | 0-19 | 20 | all | 0-15 | 16-20 | all |
| EWC [7] | - | - | - | 0.30 | 4.30 | 1.30 | 23.20 | 16.00 | 22.90 | 26.70 | 37.70 | 29.40 |
| LwF-MC [8] | - | - | - | 4.50 | 7.00 | 5.20 | 63.00 | 13.20 | 60.50 | 67.20 | 41.20 | 60.70 |
| ILT [9] | - | - | - | 3.70 | 5.70 | 4.20 | 69.10 | 16.40 | 66.40 | 63.20 | 39.50 | 57.30 |
| MiB [3] | - | - | - | 46.20 | 12.90 | 37.90 | 69.60 | 25.60 | 67.40 | 71.80 | 43.30 | 64.70 |
| PLOP [5] | - | - | - | 57.86 | 13.67 | 46.48 | 75.37 | **38.89** | 73.64 | 71.00 | 42.82 | 64.29 |
| SSUL | **65.39** | **34.90** | **50.87** | 73.97 | 32.15 | 64.01 | 77.38 | 22.43 | 74.76 | **76.44** | **45.60** | 69.10 |
| SSUL-M | **65.02** | **40.82** | **53.50** | **76.46** | **43.37** | **68.58** | **77.58** | 43.89 | **75.98** | **76.47** | **48.55** | **69.83** |
| Joint | 78.41 | 76.35 | 77.43 | 79.77 | 72.35 | 77.43 | 77.51 | 77.04 | 77.43 | 79.77 | 72.35 | 77.43 |

## 3.6 The details of experimental results of Pascal VOC 2012

Table 6 shows the summarized results of Pascal VOC 2012 by each class.

Table 6: Details of Pascal VOC 2012 mIoU performance per class.

| | bg | aero | bike | bird | boat | bottle | bus | car | cat | chair | cow | table | dog | horse | mbike | person | plant | sheep | sofa | train | tv | mIoU |
|---|---|---|---|---|---|---|---|---|---|---|---|---|---|---|---|---|---|---|---|---|---|---|
| **10-1 (11 tasks)** | | | | | | | | | | | | | | | | | | | | | | |
| SSUL | 88.55 | 84.42 | 36.93 | 84.63 | 67.40 | 80.16 | 76.53 | 88.01 | 87.88 | 34.38 | 55.55 | 27.08 | 69.92 | 41.33 | 63.27 | 80.73 | 26.89 | 39.38 | 26.43 | 46.10 | 38.69 | 59.25 |
| SSUL-M | 88.19 | 85.98 | 37.57 | 86.14 | 67.85 | 75.67 | 90.55 | 86.25 | 86.89 | 35.07 | 74.09 | 35.01 | 71.88 | 51.85 | 70.50 | 80.82 | 28.90 | 54.51 | 26.67 | 65.56 | 46.65 | 64.12 |
| **15-1 (6 tasks)** | | | | | | | | | | | | | | | | | | | | | | |
| SSUL | 89.55 | 88.99 | 41.01 | 88.37 | 69.45 | 80.82 | 85.55 | 88.47 | 92.55 | 35.52 | 77.02 | 56.74 | 90.11 | 83.46 | 84.30 | 85.12 | 32.98 | 49.63 | 26.60 | 43.38 | 30.39 | 67.61 |
| SSUL-M | 90.83 | 89.78 | 40.26 | 89.79 | 71.44 | 79.53 | 93.44 | 87.16 | 92.84 | 35.43 | 83.86 | 55.51 | 89.85 | 83.63 | 85.36 | 85.03 | 31.76 | 63.90 | 29.97 | 69.29 | 50.15 | 71.37 |
| **5-3 (6 tasks)** | | | | | | | | | | | | | | | | | | | | | | |
| SSUL | 86.49 | 73.10 | 37.84 | 85.10 | 65.05 | 79.49 | 41.21 | 59.68 | 67.67 | 12.58 | 43.94 | 37.13 | 61.67 | 35.69 | 61.22 | 78.54 | 35.61 | 46.74 | 21.00 | 34.18 | 43.85 | 52.75 |
| SSUL-M | 88.35 | 80.21 | 37.13 | 84.98 | 66.68 | 80.12 | 58.45 | 64.79 | 66.72 | 14.45 | 48.51 | 38.88 | 61.87 | 33.32 | 65.88 | 77.90 | 33.54 | 46.96 | 24.77 | 50.02 | 49.31 | 55.85 |
| **19-1 (2 tasks)** | | | | | | | | | | | | | | | | | | | | | | |
| SSUL | 92.24 | 88.93 | 40.47 | 89.59 | 70.85 | 79.60 | 95.08 | 88.61 | 93.68 | 37.25 | 84.78 | 59.76 | 89.52 | 85.88 | 87.57 | 84.61 | 61.33 | 82.25 | 54.50 | 88.14 | 29.68 | 75.44 |
| SSUL-M | 92.94 | 89.93 | 40.21 | 89.66 | 73.80 | 80.31 | 93.36 | 88.13 | 92.76 | 37.16 | 87.85 | 58.60 | 88.94 | 87.05 | 85.59 | 85.16 | 60.82 | 83.87 | 55.09 | 85.44 | 49.76 | 76.49 |
| **15-5 (2 tasks)** | | | | | | | | | | | | | | | | | | | | | | |
| SSUL | 91.54 | 89.80 | 40.01 | 88.59 | 70.28 | 81.18 | 89.66 | 88.02 | 92.81 | 36.78 | 75.70 | 56.69 | 90.00 | 83.56 | 85.17 | 85.42 | 36.03 | 57.74 | 32.08 | 70.09 | 54.57 | 71.22 |
| SSUL-M | 91.68 | 89.33 | 40.16 | 88.62 | 70.95 | 80.39 | 93.94 | 86.66 | 92.98 | 36.73 | 84.50 | 55.34 | 89.83 | 83.15 | 84.99 | 85.16 | 38.93 | 71.38 | 35.43 | 75.95 | 57.35 | 73.02 |

## 3.7 The details of experimental results of class orderings

Table 7 shows the numerical details for SSUL and SSUL-M of Figure 3(b) in the manuscript. Note that we strictly followed the class orderings of Pascal VOC 2012 as done in PLOP [5]:

Table 7: Details of the performances for each class ordering on VOC 15-1 scenario.

| 15-1 (6 tasks) | Class Ordering | | | | | | | | | | | | | | | | | | | |
|---|---|---|---|---|---|---|---|---|---|---|---|---|---|---|---|---|---|---|---|---|
| | 1 | 2 | 3 | 4 | 5 | 6 | 7 | 8 | 9 | 10 | 11 | 12 | 13 | 14 | 15 | 16 | 17 | 18 | 19 | 20 |
| **SSUL** | | | | | | | | | | | | | | | | | | | | |
| 0-15 | 77.31 | 75.48 | 72.82 | 76.45 | 77.13 | 72.39 | 76.66 | 81.43 | 74.61 | 73.98 | 77.41 | 79.76 | 78.42 | 78.47 | 73.66 | 78.42 | 75.11 | 74.36 | 74.29 | 72.19 |
| 16-20 | 36.59 | 51.37 | 53.69 | 54.70 | 47.08 | 50.99 | 40.70 | 29.67 | 49.09 | 47.94 | 38.89 | 44.86 | 49.07 | 40.22 | 50.90 | 44.75 | 50.34 | 55.24 | 47.65 | 61.40 |
| all | 67.61 | 69.74 | 68.26 | 71.27 | 69.98 | 67.29 | 68.10 | 69.10 | 68.53 | 67.78 | 68.24 | 71.45 | 71.44 | 69.37 | 68.24 | 70.41 | 69.21 | 69.81 | 67.95 | 69.62 |
| **SSUL-M** | | | | | | | | | | | | | | | | | | | | |
| 0-15 | 78.36 | 75.44 | 73.01 | 76.42 | 77.12 | 72.75 | 77.17 | 77.41 | 74.85 | 73.68 | 77.41 | 79.83 | 77.66 | 78.87 | 74.72 | 78.27 | 75.61 | 74.65 | 72.81 | 70.85 |
| 16-20 | 49.01 | 53.99 | 72.01 | 68.77 | 58.78 | 65.58 | 55.74 | 52.88 | 57.09 | 55.45 | 52.88 | 48.33 | 55.13 | 50.22 | 64.42 | 54.00 | 63.46 | 62.55 | 47.93 | 61.27 |
| all | 71.37 | 70.33 | 72.77 | 74.60 | 72.75 | 71.01 | 72.07 | 71.57 | 70.62 | 69.34 | 71.57 | 72.33 | 72.29 | 72.05 | 72.27 | 72.49 | 72.72 | 71.77 | 66.89 | 68.57 |

## 4 Qualitative analysis on ADE20K

In Figure 1, we visualized the qualitative results from ADE20K 100-10 (6 tasks) scenario. We argue that we seldom suffer from the background semantic shift issue on ADE20K because its clear and dense labels for both things and stuff. Consequently, the false-positive predictions are noticeably reduced compared to the results on Pascal VOC 2012. As in Figure 1, the *unknown* label (*i.e.*, *black* pixels) is correctly transformed to the label to be learned in the future (*e.g.*, *fan* in step-5 and *plate* in step-6) while keeping the previously learned knowledge.

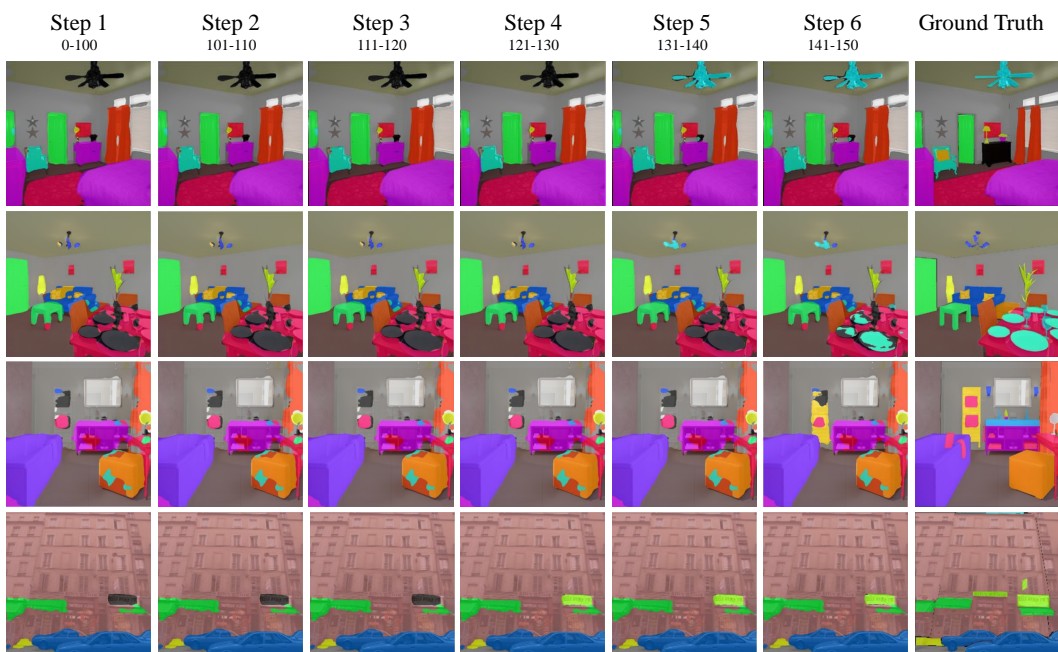

Figure 1: Qualitative results of SSUL-M on ADE20K.