# OpenReview forum: "SSUL: Semantic Segmentation with Unknown Label for Exemplar-based Class-Incremental Learning"
_NeurIPS.cc/2021/Conference — NeurIPS 2021 Poster_

### Official Review · Reviewer_zwbF · 2021-07-10

**Rating:** 5
**Confidence:** 4

**Summary:**

This paper proposes novel a class-incremental learning method, named semantic segmentation with unknown label with memory. The proposed method addresses two additional challenges in class-incremental learning and proposes three contributions: label augmentation, stable score learning, and tiny exemplar memory.  The proposed method achieves impressive performance compared with other methods.

**Limitations And Societal Impact:**

1 There are some related methods missed, such as [1][2][3].
2 The proposed method employs three strategies to alleviate catastrophic forgetting. These strategies are often employed in other incremental methods. Thus, the contributions are limited in my opinion.
3 Compared with the ground truth, the visualization results are terrible. Thus,  I want to see some analysis of satisfied cases and terrible cases.
[1] Class-incremental learning for semantic segmentation re-using neither old data nor old labels. ITSC2020.
[2] Modeling the background for incremental learning in semantic segmentation. CVPR2020.
[3] Incremental Learning Techniques for Semantic Segmentation. ICCVW2019

**Main Review:**

 This paper proposes a novel class-incremental semantic segmentation setting. In addition, the writing of this paper is clear and easy to understand. The experiments are sufficient to prove the effectiveness of the proposed paper.

**Time Spent Reviewing:**

Two hours

---

> ### Author Response · Authors · 2021-08-10
> **Rebuttal for [R4]**
>
> ## 1. Related work
> Thanks for suggesting related works. Among your suggestions, we already cited [3] and [23] of the manuscript and we will cite [1] of your suggestion as the additional related work in the final version.
>
> ## 2. Visualization results
> We are sorry that you are not satisfied with the qualitative result in Figure 4 (in the manuscript). We truly want to show some other visualizations but there is no way to add new figures during the rebuttal process. Therefore, we will add more visualizations with more analysis in the final version so please understand it.
>
> ## 3. The limited contributions
> See above 'Common rebuttal for [R1, R3, R4]: The overall novelty of this work'.

---

### Official Review · Reviewer_8KnD · 2021-07-13

**Rating:** 6
**Confidence:** 5

**Summary:**

This paper provided a recipe of methods to improve the incremental learning on semantic segmentation that empirically improved the performance on multiple datasets compared to other SOTA methods. Specifically, both pseudo labels from previous models and unknown classes from saliency detectors are adopted in model training; Backbone networks are frozen and Sigmoid + BCE are used to avoid forgetting; finally an exemplar dataset is used here to balance the training data in later times and hence further boost the performance.


**Limitations And Societal Impact:**

No negative societal impact as far I know.

The limitation of this work has not been fully discussed in the paper. Specifically, the overall method relies on the success of the saliency detector to label the unknown classes. If a new class is not able to be picked up by saliency detectors, or even the new class is simply not of “object-like” but rather “area-like”, I’d expect the proposed method to have a hard time to accommodate the new classes, especially with model freezing.

Additionally, the benefit of model freezing has a positive effect with the underlying assumption that the initial data are enough for the model to learn good features, which may not be valid in other setup of datasets. Many things can go south here, e.g. new classes are not included in the previous unknown classes, learnt feature space bounded by the initial dataset, etc.


**Main Review:**

The paper is well written in general except a few questions that may be worth considering below. The originality of the work is moderate. Adding additional label sources and/or training data, however not exact, can usually improve deep models’ performance. The paper found an interesting and valid way to do so under the incremental learning setup. Using Sigmoid + BCE is another interesting application under this setting and the explanation makes sense.  Exemplar memory throws more data into the workflow -  more often than not, it will work if doing it right.

A few question that may need further clarification:

With model freezing and stable score learning, in theory there would not be any forgetting for the foreground classes in t - 1? The explanation of having examples to avoid severe forgetting of any foreground classes seems unconvincing in Line 208.

More detailed explanation on the effectiveness of weight transfer would be helpful for the audience to appreciate the design better. In particular, after model freezing with only BCE in training, one would think that initialization matters less given the shallow structure. Why such a big difference as shown in the ablation study?

The ablation study is fascinating and considering the work is a recipe of multiple steps, it is worth more discussion and careful look to shed more light on the importance of each step. For example,

Without freezing, why also such a big drop on new classes as well?

It is argued earlier in the paper (3.4) that model freezing “can roughly learn the representations for the potential future class objects…” in the unknown class. However, in the ablation study, the model performs really well even without the unknown classes. In fact, the performance is still higher than most of the SOTA methods. What is the underlying reason here? Does the composition of data have any influence here as well?

Any chance we can show the ablation study without PL? It is helpful to understand how important it is to have PL even if the model is frozen.

It would be interesting to see the impact of each step with respect to the complexity of the problem, i.e. reduce the initial number of classes. For example, if we increase the complexity of the problem, is there a certain point that model freezing starts to hurt the performance?



**Time Spent Reviewing:**

1.5 hrs

---

> ### Author Response · Authors · 2021-08-10
> **Rebuttal for [R3]**
>
> ## 1. Freezing and forgetting
>
> We apologize for not clearly explaining in line 208-213. What we meant was that even with model freezing and stable score learning, when the confidence for the new class is learned to be high for a pixel (potentially for an old class), the prediction for the pixel could get biased toward the new class, causing the forgetting of the old class. As described in line 208-213, this could happen when the given image never contains object cues for the old classes. But, when we use exemplar memory, it helps the model not become overconfident of new classes for the pixel for old classes; hence, it can give significant boost of performance for both old and new classes. In the final version, we will revise line 208-213 with a clear explanation.
>
> ## 2. Weight transfer
>
> Thanks for the insightful comment and the question. One may think the initialization of the final linear layer would not matter since the feature extractor is frozen. However, as also observed in prior work on CIL [1][12], the number of epochs or the level of overfitting during learning seems to play an important role in finding the right trade-off between learning new classes and not forgetting the old classes.
>
> To that regard, we did additional experiments checking the effect of initialization and the number of epochs. Namely, we simply did the random initialization for $\phi_c^t$ and increased the number of epochs (x1, x2, x4) and compared with our weight transfer (x1) results. From the table, we observe that the random initialization cannot reach the performance of weight transfer even with a larger number of epochs.
>
> We argue this is a similar phenomenon as what we observe in ordinary CIL. Namely, for the new class sigmoid, it is true that the linear weight would converge to the same solution regardless of the initialization. However, the number of epochs to reach the solution would differ -- if we do the weight transfer, it will converge quicker, whereas if randomly initialized, a larger epoch would be necessary. Now, in the latter case, running with a larger epoch would result in the overfitting of the weights for $c_u$ and $c_b$ to the noisy pseudo label.  Then, it will cause overconfident predictions for the Unknown or Background classes, which would penalize the overall mIoU. On the other hand, when only a small  number of epochs are used for the random initialization, the prediction for the new class would become inaccurate, which again would hurt the overall mIoU.
>
> |Head Init|0-15|16-20|all|
> |:-:|:-:|:-:|:-:|
> |Rand x1|73.73|23.99|61.89|
> |Rand x2|73.89|23.08|62.45|
> |Rand x4|72.22|22.38|61.14|
> |Weight Transfer x1|78.06|28.54|66.27|
>
> ## 3. Only freezing surpasses SOTA
> See above ‘Common rebuttal for [R1, R3]: On model freezing and plasticity.’
>
> ## 4. Without PL performance
> We experimented on the w/o PL case, following the reviewer’s suggestion. By comparing SSUL with SSUL (w/o PL), we checked that PL leads to an increase of 2.1% of the mIoU. We will add this result to ouron our final version.
>
> ## 5. Reducing the initial number of classes
> See above official comment on ‘Common rebuttal for [R1,R3]: On model freezing and plasticity’
>
> ## 6. Limitations
> Thanks for the suggestion. Thanks to all the reviewer's helpful comments, we think we should describe our limitations and future works.
> First, our unknown class modeling may not be helpful for some stuff segmentation tasks because saliency maps are mainly targeting things  (or objects) (see above response).
> Second, our model freezing may suffer from learning new classes that are not captured as the unknown class in the first step; it needs to be discussed with some failure cases.
> We will clearly specify these limitations in the final version.

---

> > ### Comment · Reviewer_8KnD · 2021-09-01
> > **Post-rebuttal comment**
> >
> > Thank you for the response. Based on other reviews and comments, I inclined to keep my rating as "Marginally above the acceptance threshold".

---

### Official Review · Reviewer_xEzA · 2021-07-15

**Rating:** 6
**Confidence:** 3

**Summary:**

This paper proposes a method, Semantic Segmentation with Unknown Label with Memory (SSUL-M), to solve a class-incremental semantic segmentation (CISS) problem. The authors make three main contributions. First, they introduce an additional “Unknown” class label to distinguish the potential future class and the true background. Second, they adopt the pseudo-labeling strategy and freeze backbone network and past classiﬁers with binary cross-entropy loss to tackle catastrophic forgetting. Third, an exemplar memory is utilized to improve both plasticity and stability. Extensive experiments and ablation studies are conducted to justify these proposed components and contributions. Results show that SSUL-M signiﬁcantly outperformed baselines.

**Limitations And Societal Impact:**

The authors did not describe the limitations in the paper and stated no expected potential negative societal impact.

**Main Review:**

1.	This paper is well structured and clearly written. The authors introduce the proposed method logically and describe their contributions clearly. The experimental details are provided and look technically sound and reproducible.
2.	This work aims to tackle the challenges that cause catastrophic forgetting in CISS. To this end, the authors make three main contributions which are quite novel and each contribution has been proved effective by adequate ablation studies. Experimental results on two datasets show that the proposed method achieves signiﬁcantly better performance than the baselines.
3.	Figure 1 is not illustrated clearly. It would be better if the authors explain this figure clearly in its caption so that readers can quickly understand the meaning of each module or symbol in this figure at a glance. Additionally, there are some typos and grammar mistakes. I list several of them here: “SSUL and SSUL-M both maintains … and effectively learns” in page 8, “The ﬁrst two columns in Table 4 shows” in page 9, etc.


**Time Spent Reviewing:**

5H

---

> ### Author Response · Authors · 2021-08-07
> **Rebuttal for [R2]**
>
> Thanks for the positive review for our paper. We will add more explanations for Figure 1 and correct types and grammar mistakes in the final version.

---

> > ### Comment · Reviewer_xEzA · 2021-08-25
> > **Post-Rebuttal Comment**
> >
> > Thanks for providing the responses. I also read the other reviewers' comments and would like to keep my original rating.

---

### Official Review · Reviewer_eHGE · 2021-07-18

**Rating:** 6
**Confidence:** 4

**Summary:**

The paper presents a class-incremental learning strategy for semantic segmentation. The main contributions of this work include two aspects: first, it introduces an external saliency map predictor to better model the thing classes vs the stuff background; second, the method adds a class-balanced memory to overcome the forgetting in CIL. The proposed method is evaluated on two benchmarks with the overlapped setup and achieves better performance than the prior approaches.

**Limitations And Societal Impact:**

Should add discussion on its limitations.

**Main Review:**

Pros:
- The paper developed a well-engineered strategy for class incremental segmentation, which integrates several well-motivated components to reduce catastrophic forgetting.
- The proposed method achieves strong results on two benchmarks under the overlapped setup, which validates its efficacy. It also includes a detailed ablative study on the main components.
- The paper is most clearly written and easy to follow.


Cons:
- The overall novelty of this work is a bit incremental. While the main components are clearly motivated, they are largely borrowed from the prior work in weakly-supervised segmentation (foreground saliency), class-incremental learning (memory), and [7] (pseudo-labels). The beneficial effects of those modules seem to be well understood.

- There are several concerns on the technical aspects of the method:
1) While adopting object saliency maps can help the thing classes, this would have a negative impact on the stuff classes. For semantic segmentation, this is restrictive for the general cases where we have to incrementally segment both the thing and stuff classes. This also implicitly introduces additional training data (saliency) which may not available for certain application domains.
2) The model freezing step is also rather restrictive, which leads to weak plasticity for the model. In particular, the performance of the convolutional feature extractor can degrade significantly if the initial class set is small. It is unclear how well such a strategy can generalize to different CIL setups.
3) How are the hyperparameters of the method determined, such as \tau in pseudo labeling?
4) Line 189: if the sigmoid output is adopted in training, how does this method calibrate the classifier scores for the inference in Sec 3.1?

- The experimental evaluation seems a bit lacking and the conclusions are not fully convincing:
1) The evaluation is only conducted on the overlapped setup, which is insufficient. The method should be tested on the disjoint setup too. The overlapped setup allows the method to learn features for future classes based on the saliency mask, which would favor the proposed strategy.
2) How well the method would perform if the classes include both thing and stuff categories?
3) Using external saliency maps seems to be unfair to other methods, as it requires additional training data. Such additional information can also help improve the prior methods.
4) In the ablative study, it seems that adding the Sigmoid+BCE hurts the overall performance. It is unclear if this is a necessary component as Sec 3.4 claims.

Post-rebuttal: The author's reply addressed many of my initial concerns and I would like to raise my rating accordingly.

**Time Spent Reviewing:**

2~3

---

> ### Author Response · Authors · 2021-08-10
> **Rebuttal for [R1]**
>
> ## 1. Saliency map
>
> Thanks for the insightful review on the saliency detector. In order to observe the effect of using the saliency-detector on the performance of other baselines, we additionally experimented with a variant of PLOP. That is, we implemented PLOP (refine), which only used the pseudo-labels that exist in the salient regions, similarly as in ours. As we can see in the table below, PLOP (refine) slightly improves the original PLOP by 2.37% (i.e., 54.64 → 57.01). However, such result is still significantly lower than the result of SSUL (66.27%), underscoring the fact that our gain is not merely stemming from the additional data required for training the saliency detector.
>
> Furthermore, even in our SSUL, we note that just naively using the saliency map would not automatically bring the mIoU gain. Namely, the saliency detector is closely used with our unknown class modeling, however, as we can see in Table 3 (2nd line) and Table 4 (weight transfer column), if the correct weight transfer is not used, the mIoU becomes 61.89% even with the unknown class, which is not much higher than the result without the saliency detector and unknown class, i.e., 61.12%. Note we can enjoy the improvement to 66.27% (+5.15%) only when the proper weight transfer is applied.
>
> With the above, we argue that our performance gain is not merely from utilizing more data but from the careful combination of several right techniques, which are far from straightforward.
>
> ||Saliency|0-15|16-20|all|
> |:-:|:-:|:-:|:-:|:-:|
> |PLOP|X|65.12|21.11|54.64|
> |PLOP|O (refine)|67.25|24.22|57.01|
> |SSUL|O|78.06|28.54|66.27|
>
> ## 2.  Freezing and plasticity, small initial class set
> see ‘Common rebuttal for [R1,R3]: On model freezing and plasticity’
>
> ## 3. How determine the hyperparameters such as \tau
> The \tau controls the reliability of the pseudo-labels and we simply set the value to 0.5 without extensive tuning (as we mentioned in line 158). It turns out the final performance is not very sensitive to \tau, i.e., only within 1% performance variation, and we will mention and include such results in the camera-ready version.
>
> ## 4. Calibrate the classifier scores
> For the confidence map, we apply the sigmoid activation function on the last feature maps of the network f (see L.157). During training, we apply the pixel-wise BCE loss function on the confidence map and ground truth (one-hot) map, and note this strategy is already used in Mask R-CNN for instance segmentation.
>
> ## 5. Disjoint setup
> We would like to argue that the reason we only take the overlapped setup is not because of the favor of our method, but because it is more challenging and practical. As mentioned in previous papers (MiB and PLOP), the disjoint setup has an additional constraint that only the classes observed so far are present in the input image, which is not necessarily practical.
>
> By following the reviewer’s suggestion, we did an additional experiment on the disjoint setup as well as shown in the following  table. We clearly observe that SSUL and SSUL-M again achieve superior results compared to other baselines, which shows that our method is not particularly favoring the overlapped setup. We will add this result in our final version as well.
>
> |Method|0-15|16-20|all|
> |:-:|:-:|:-:|:-:|
> |MiB|46.20|12.90|37.90|
> |PLOP|57.86|13.67|46.48|
> |SSUL|75.71|22.96|63.15|
> |SSUL-M|75.75|29.69|64.78|
>
> ## 6. Thing and stuff classes
> see ‘Common rebuttal for [R1, R3]: The case for stuff (area-like) and thing (object-like) classes in CISS’
>
> ## 7. Sigmoid + BCE hurts the overall performance
> We believe that the reviewer misunderstood Table 3 in the manuscript. This table shows that Sigmoid+BCE significantly improves (not hurts) the overall performance (compare 1st and 4th line of the paper).

---

> > ### Comment · Reviewer_eHGE · 2021-09-01
> > **Post rebuttal**
> >
> > I thank the author's reply to my initial comments. Many of my original concerns have been well addressed and the additional experimental results provided further evidence to support the paper's claims. That said, I am not fully convinced by adopting the saliency map and the model freezing strategy - their empirical benefits might be due to the data bias. Overall, considering its strong benchmark performance, I would like to raise my rating to the positive side.

---

### Author Response · Authors · 2021-08-10
**Common rebuttals**

We first designate the reviewer numbers as follows.
- R1: Reviewer eHGE
- R3: Reviewer 8KnD
- R4: Reviewer zwbF

## Common rebuttal for [R1, R3, R4]: The overall novelty of this work

[R1, R4] We respectfully disagree that our contribution is incremental. Even though each component we use --- introducing unknown class, stable score learning (model freezing + sigmoid), and exemplar-memory --- may seem common, we argue that carefully combining them with right intuition for the CISS problem and achieving significantly better performance over the state-of-the-arts is far from being straightforward.

[R1, R3] Particularly, we tailored our method to address the two novel challenges that the CISS problem possesses, which are clearly elaborated in Section 3.2 (line 121-139). While the first challenge on the semantic drift of the background (BG) labelBG shift has been also considered in other work, the second challenge that stems from the nature of multi-label prediction in CISS has been largely neglected in previous work. Namely, since the segmentation model should correctly predict the class label for each pixel, the segmentation performance (measured with mIoU) for each class has an effect on the performance of other classes as well. This is why maintaining the mIoU for the old classes is extremely important for improving the mIoU for the newly learned classes since it can reduce the false positives for the new classes (see line 129-139 for more details).

We tackled the first challenge by introducing the unknown class label, and identifying the second challenge is precisely why we introduced our stable score learning (model freezing+sigmoid) and exemplar-memory. While all the previous work simply focused on the first challenge, we found in our work that addressing the second challenge is also very important for achieving high performance as can be seen in our ablation study (Table 3) and more discussions below.

[R1, R4] We believe a novelty of a paper can also lie in identifying a neglected, novel challenge of a considered problem and providing a carefully implemented solution to achieve significantly better performance than the closest baseline (e.g., 32 mIoU improvement (relative 105% improvement)  over PLOP for VOC 10-1 task in Table 1). We sincerely hope the reviewers can re-evaluate the novelty of our paper based on our rebuttal.


## Common rebuttal for [R1,R3]: On model freezing and plasticity

[R1, R3] As the reviewers have pointed out, the model freezing may seem too restrictive as it may focus too much on maintaining the performance of the old classes and limit the capability of learning new classes. However, in our ablation study (Table 3), by comparing 1st and 3rd line of the table, we observe that model freezing significantly improves the mIoU not only for the old classes, but also for the new classes. This somewhat counterintuitive phenomenon mainly has to do with the second challenge mentioned above --- i.e, since the predictions for the old classes remain accurate, the mIoU for the new classes also improves by limiting the false positives.

With such a positive effect of model freezing on the mIoU for the new classes, the unknown class label is introduced to compensate for the restrictiveness of complete freezing of the feature extractor in learning new classes and deal with the BG shift issue. Comparing the 1st and 2nd line of Table 3 shows that the unknown class clearly further improves the performance for the new classes by effectively learning the representations of potential future classes (embedded in the unknown class). Note the result of SSUL (for VOC 15-1 task) is better than that of PLOP not only for the old classes, but also for the new classes when the unknown class label is used.

[R1] The result for the case in which the initial class set is small can be found in Table 1 (VOC 5-3 task), and we observe our method still significantly surpasses the baselines. Furthermore, we carried out additional experiments with small initial classes (i.e., VOC 5-1 and 2-1 tasks) and summarized their results together with other single class incremental settings (VOC 10-1,15-1,19-1 tasks) below. We again observe that our SSUL consistently outperforms other baselines by a large margin even for such extreme cases, which confirms the effectiveness of our SSUL method.

|Task|SSUL-M|SSUL|PLOP|MiB|
|:-:|:-:|:-:|:-:|:-:|
|19-1|76.71|76.07|73.54|69.15|
|15-1|70.58|66.27|54.64|29.29|
|10-1|62.45|58.23|30.45|12.65|
|5-1|49.66|43.30|6.46|10.03|
|2-1|38.49|30.93|4.47|9.88|

[R1, R3] Finally, the learning capability for the new classes significantly gets boosted with the usage of the exemplar-memory. That is, for the VOC 15-1 task in Table 1, the mIoU for the new classes improves from 28.54 (SSUL) to 43.86 (SSUL-M). This again is due to the fact that the ground-truth labels for the old classes in the memory help make less false positive errors for the new classes.

In summary, although the model freezing may seem somewhat counterintuitive for achieving the plasticity, we convincingly show that it is very effective not only for maintaining the performance of the old classes, but also for learning the new classes in the CISS problem.


## Common rebuttal for [R1, R3]: The case for stuff (area-like) and thing (object-like) classes in CISS

One of our main target issues is the semantic drift between true background and future class (line 121), and we alleviate the issue using the unknown class modeling with saliency maps. We believe that the applicability of the saliency maps depends on the dataset condition, that is, the matter is how many true backgrounds are included in the dataset.

Firstly, the segmentation datasets including only the object/thing (e.g., Pascal VOC) contain a huge number of true backgrounds. In this case, we showed that our unknown modeling with saliency maps is greatly helpful to alleviate the semantic drift between true background and future class (in Table 1, Figure 3 and Figure 4).

Secondly, if the segmentation dataset including both the thing and stuff contains few true background pixels (e.g., ADE20K), the semantic drift rarely occurs (line 301). In this case, we can apply our unknown modeling without saliency maps (see line 300-308) and achieve a reasonable performance as in Table 2.

Finally, if the segmentation dataset including the thing and stuff contains many true backgrounds, the saliency maps may not be helpful as concerns [R1, R3]. We concede it is our limitation and will add the section for discussing this limitation in our final version.

Nevertheless, we respectfully argue that the final case, that is, the dataset supports stuff segmentation yet contains many true backgrounds is not a common condition.  Most famous datasets (e.g., Cityscape, COCO-stuff, Pascal Context, and NYUD) contain few true backgrounds, therefore, our unknown modeling can widely be utilized in the same way as applied to ADE20K.

---

### Decision · Program_Chairs · 2021-09-27

**Decision:**

Accept (Poster)

**Comment:**

The paper describes an effective method for class-incremental semantic segmentation.
The different components integrated to achieve this goal have limited novelty per se,
but are combined nicely and result in a system that works well, particularly under the non-disjoint setting.
Authors are encouraged to include a discussion on limitations in the final version of the paper.